# Analysis of Serial Neuroblastoma PDX Passages in Mice Allows the Identification of New Mediators of Neuroblastoma Aggressiveness

**DOI:** 10.3390/ijms24021590

**Published:** 2023-01-13

**Authors:** María A. Gómez-Muñoz, Diana Aguilar-Morante, Ana Colmenero-Repiso, Aida Amador-Álvarez, Mónica Ojeda-Puertas, Juan Antonio Cordero Varela, Ismael Rodríguez-Prieto, Ricardo Pardal, Francisco M. Vega

**Affiliations:** 1Medical Physiology and Biophysics Department, Universidad de Sevilla and Instituto de Biomedicina de Sevilla (IBiS) (Hospital Universitario Virgen del Rocío/CSIC/Universidad de Sevilla), 41013 Seville, Spain; 2Cell Biology Department, Faculty of Biology, Universidad de Sevilla and Instituto de Biomedicina de Sevilla (IBiS) (Hospital Universitario Virgen del Rocío/CSIC/Universidad de Sevilla), 41013 Seville, Spain; 3Bioinformatics and Computational Biology Service, Instituto de Biomedicina de Sevilla (IBiS) (Hospital Universitario Virgen del Rocío/CSIC/Universidad de Sevilla), 41013 Seville, Spain

**Keywords:** neuroblastoma, PDX, gene expression, miRNA, differentiation

## Abstract

Neuroblastoma is a neural crest cell-derived pediatric tumor characterized by high inter- and intra-tumor heterogeneity, and by a poor outcome in advanced stages. Patient-derived xenografts (PDXs) have been shown to be useful models for preserving and expanding original patient biopsies in vivo, and for studying neuroblastoma biology in a more physiological setting. The maintenance of genetic, histologic, and phenotypic characteristics of the original biopsy along serial PDX passages in mice is a major concern regarding this model. Here we analyze consecutive PDX passages in mice, at both transcriptomic and histological levels, in order to identify potential changes or highlight similarities to the primary sample. We studied temporal changes using mRNA and miRNA expression and correlate those with neuroblastoma aggressiveness using patient-derived databases. We observed a shortening of tumor onset and an increase in proliferative potential in the PDXs along serial passages. This behavior correlates with changes in the expression of genes related to cell proliferation and neuronal differentiation, including signaling pathways described as relevant for neuroblastoma malignancy. We also identified new genes and miRNAs that can be used to stratify patients according to survival, and which could be potential new players in neuroblastoma aggressiveness. Our results highlight the usefulness of the PDX neuroblastoma model and reflect phenotypic changes that might be occurring in the mouse environment. These findings could be useful for understanding the progression of tumor aggressiveness in this pathology.

## 1. Introduction

Neuroblastoma (NB) is the most common extracranial solid tumor occurring in childhood and it is characterized by poor outcomes in advanced stages, with a high frequency of metastasis, relapse, and resistance to therapy [1]. NB originates during sympatho-adrenergic differentiation from the neural crest [2]. Aberrations during neural crest cell migration and/or differentiation processes within the sympatho-adrenal lineage are the main causes for neuroblastoma initiation and progression. Since the neural crest is particularly plastic, neuroblastoma tumors are characterized by a high heterogeneity [3].

The differentiation status of the tumor tissue has been shown to play an important part in the definition of aggressiveness and prognosis [4]. Recent transcriptomic and single-cell technology studies show that high-risk tumors present a greater enrichment of cells with an undifferentiated nature, including a subpopulation of neural crest stem-like progenitor cells associated with malignancy [5,6,7]. Tumors with amplification of the oncogene MYCN, a common molecular feature in high-risk tumors, present an enrichment in early and proliferative neuroblasts, while non-amplified MYCN tumors are characterized by an enrichment in late, or more differentiated, neuroblasts [8]. In fact, higher expression of genes associated with immature sympathoblasts correlates with worse patient survival [7], whereas mainly differentiated neuroblasts are found in low-risk tumors.

The study of neuroblastoma has classically relied on data provided by immortalized cancer cell lines. However, to study neuroblastoma disease in vivo, several mouse models have been developed through genetic engineering [9,10], the TH-MYCN transgenic mouse model (which expresses human MYCN specifically in neuroblasts under the control of the tyrosine hydroxylase (TH) promoter) being the most widely used. All these models develop spontaneous neuroblastoma tumors. However, although genetically engineered mouse models are useful in vivo models for studying neuroblastoma, they fail to represent the enormous clinical heterogeneity found in the human disease. For this reason, patient-derived xenografts (PDXs) in immunocompromised mice have been developed [11]. PDXs avoid cell selection and adaptation typical of in vitro conditions, hence allowing a more comprehensive representation of the diversity of high-risk neuroblastomas. Moreover, this model provides opportunities to refine personalized therapy recommendations for neuroblastoma patients [12,13].

Neuroblastoma PDXs have previously been compared to their corresponding original tumors by performing histopathological characterization and whole-genome genotyping array analyses [14]. These studies confirm that PDXs retain the differentiation status of their corresponding patient tumors and the clinical features of the disease, such as the protein marker profile. It has also been shown that PDX tumors retain the original chromosomal aberrations and are able to form distant metastases in mice, appearing as robust models for invasive NB [15,16]. All these neuroblastoma features are supposed to be preserved in the PDXs through serial in vivo passaging [17]. Nevertheless, some new mutations that are not present in the original tumors have been found in their corresponding PDXs. It has been postulated that these mutations could come from a minor clone in a distinct region of the primary patient tumor that has a growth advantage in the xenograft or that they could be acquired de novo in the xenograft with passages [16]. Therefore, although largely resembling the original tumor biopsy sample, PDXs could eventually show phenotypic differences that could be reflected in new gene expression patterns.

In this manuscript, we analyze the behavior of neuroblastoma PDXs in serial passages in immunocompromised mice. We show that PDX tumors progressively turn more proliferative and aggressive. In order to characterize this development, we performed extensive mRNA and miRNA expression bioinformatic analysis. We postulate that the increase in aggressiveness of these PDX tumors can be exploited to explore gene expression changes related to malignancy and to identify new genes involved in neuroblastoma progression.

## 2. Results

### 2.1. Successive PDX Passages in Mice Show a Pattern of Increased Aggressiveness

Patient-derived xenograft models are being successfully used for a variety of solid tumors and have been described as recapitulating human tumor architecture [11]. In neuroblastoma, PDXs have also been shown to serve as a source of tumor material and as a surrogate for scarce human tumor samples, maintaining similar genetic characteristics over time to the original patient tumor sample. In the laboratory, we developed several PDX models from freshly obtained neuroblastoma patient tumor biopsies (Appendix A). These PDXs were serially maintained on immunocompromised mice according to the described protocol (Figure 1a). The engraftment rate of fresh patient samples in the SCID immunocompromised mice used was 10.2%. We consistently observed a decrease in the time it took the mice to reach the point of sacrifice with increased PDX passages, proportional to the time needed for engraftment (Figure 1b). Some authors have proposed a necessary adaptation of the human tumor to the mouse environment in which human stroma is sequentially substituted by one provided by the host. We wanted to explore whether this adaptation correlates with an increase in aggressiveness and with changes in the differentiation status of the tumor by analyzing gene expression changes among serial PDX engraftments over time. We obtained mRNA from PDX NB27T tumors after the first engraftment in mice (PDX1) and from tumors on subsequent PDX passages (PDX 2 to 5), and performed microarray gene expression analysis and miRNA analysis of these samples. From the global expression profiles, we could observe that, although there were some changes associated with PDX passages, these were not very drastic, and overall, mRNA or miRNA expression profiles remained similar after 5 successive passages in mice (Figure 1c,d). This result highlights the usefulness of the PDX model as a tool to explore human solid tumor behavior and amplify tumor biopsy samples. A stepwise analysis for the mRNA expression showed a greater number of changes in the intermediate passages (PDX2-PDX3), but an accumulative effect on gene expression changes that increased with each passage from the reference PDX1 up to the PDX5 (Appendix A).

Despite the absence of drastic changes in expression, we set out to explore if there were minor changes in signaling networks that could be associated with changes in the differentiation or proliferative status of the tumor and its aggressiveness and that might also explain the observed differences in survival.

### 2.2. PDX Gene Expression Patterns over Time Associate with Changes in Undifferentiation and Proliferation Genes

To correctly analyze the changes associated with successive passages of PDX NB27T in mice, we performed an analysis with Short Time-series Expression Miner (STEM, v1.3.13) software [18], which groups genes according to their changing patterns over time. We focused on the patterns that were significantly represented in the dataset, with a significant number of genes with representative *p* values. We could observe genes with increased expression patterns, genes with decreased expression patterns, and mixed patterns in which there were upregulations and downregulations over time (Figure 2a). We also performed gene ontology analysis and observed that the most represented biological processes in the increasing patterns were related to regulation of cell proliferation, neuronal development and differentiation, and cell migration, while differentiation and cell matrix regulation were the ones most represented in the decreasing patterns (Appendix A). Interestingly, biological functions related to cytokine regulation were among the decreasing ones. Among the genes following an increasing pattern over passages were genes involved in neural function (such as OLFM1, SRGAP1, or NLGN1), in matrix microenvironments (such as laminin B1, NRP2, or SEMA5A), or progenitor markers (such as LGR5). Among the ones decreasing their expression over the passages were neurexin1, NGF, IL-7, or the collagen gene COL1A2 (Appendix A).

To better identify single genes whose expression was significantly changing from PDX1 to PDX5 we compared the profiles from these two isolated conditions. In general, changes obtained in gene expression were not greater than twofold, but the differences between samples were significant. Volcano plots show that there were some differentially expressed genes on PDX5 compared to PDX1, including genes involved in proliferation or differentiation, such as ARHGAP20, CNBD1, PLCE1, DKK1, or ADAMTS9 (Figure 2b and Appendix A). We also detected significant changes in some miRNAs, including some that have been related to the control of cell proliferation and differentiation, such as miR429, miR374C, miR758, miR376B, and miR329-2, and some involved in neuroblastoma progression, such as miR3167.

Lastly, we explored different gene signatures from published single-cell gene expression datasets to try to identify cell types or biological processes that were enriched in PDX NB27T, passage 5, compared to the first passages (Figure 3). We could see a drop in mesenchymal differentiated cells and stromal components and an increase in proliferative neuroblasts and committed progenitors, among others, along serial passages in mice. We could also detect an increase in unfavorable signatures and a decrease in favorable marker genes. Interestingly, we observed an increase in mesenchymal signature genes, such as the adhesion molecule CD44, the PDGF receptor beta, or PPRX, and an increase in the BRCA tumor suppressor and a decrease in some classical progenitor markers, such as prominin-1 or CSMD3. No significant changes were observed in the miRNA signature described as being associated with high risk in neuroblastoma [19] (Appendix A).

### 2.3. miRNA for Relevant Neuroblastoma Targets Are Associated with PDX Passage

mRNA expression analyses in tumor samples are common. Less attention has been paid to miRNA regulation of gene expression. As we disposed of parallel miRNA and mRNA expression data, we decided to look for possible changes in regulatory networks involving miRNAs in our dataset, which could be responsible for the phenotype. Using a combination of ANOVA and Kruskal–Wallis analysis of the data, we performed an interactome analysis, identifying a total of 285 miRNAs changing along the PDX passages. From those, 142 changed between PDX1 and PDX5, potentially regulating the expression of a total of 990 mRNA targets (Figure 4a). Among them, the miRNAs miR-34b-3p, miR-29a-3p, miR-34c-5p, and miR-29b-3p appear upregulated in PDX5, and one of their possible targets is the oncogene MYCN mRNA, which is highly maintained over the passages, as the original tumor is NMYC-amplified. Several of the identified miRNAs are acting over CD44 expression, which is a marker for neuroblastoma undifferentiated cells, and is also present in glial derivatives. Among regulated miRNAs, miR-130b-5p appears to be significantly downregulated in PDX5. This microRNA seems to negatively regulate CD44 and positively regulate the progenitor marker prominin-1 (CD133), which are respectively upregulated and downregulated in PDX5 versus PDX1 (Figure 4b). Another significantly affected miRNA is miR-146b-5p, which modifies the expression of the progenitor and proliferative protein c-KIT.

To elucidate the relevance of these changing miRNAs in neuroblastoma progression, we explored the correlation with patient survival of all annotated miRNAs with significant changes, either individually or all together, as a miRNA expression signature. We were able to identify an unknown association of miRNA-99a with neuroblastoma progression. miRNA-99a was significantly upregulated in PDX5 versus PDX1 and significantly correlated to MYCN amplification and poor outcome and survival, with a high expression associated with worse survival (Figure 4c,d). miRNA-99a has been involved in the negative regulation of IL-6-mediated signaling and the receptor signaling pathway via STAT, although specific targets have not been identified.

### 2.4. PDX Passages Permit the Identification of Genes with Relevance in Neuroblastoma Progression

Bioinformatic mRNA gene expression analysis and miRNA interactome analysis have shown, thus far, that there are expression changes associated with PDX passages in mice and that those changes are partially related to genes and pathways involved in cell proliferation and differentiation in neuroblastoma. To demonstrate that the differences in PDX tumor onset and observed aggressiveness are concordant with a different tumor phenotype, we performed immunohistochemistry on tumors from the different PDX passages using common markers for neuroblastoma differentiation and proliferation (Figure 5a,b). Tumors presented an increase in Ki67 proliferation marker and a decrease in DDC (dopamine decarboxylase), a marker for sympatho-adrenal differentiation, along passages from PDX1 to PDX5. There was also a significant increase in the proliferation marker VRK1.

We also wanted to identify whether the genes changing in NB PDXs with passages could be involved in neuroblastoma progression. With this aim, we selected the top 20 genes changing the most between PDX1 and PDX5 and analyzed whether the expression of these genes was beneficial or detrimental for neuroblastoma patient survival in various tumor datasets. We identified a subset of genes upregulated in PDX5 over PDX1 whose over-expression in patient tumors consistently associates with low survival (Figure 5c and Appendix A). Those genes were SLC1A5, SHFM1, LSM2, and WDR46. Conversely, we identified genes downregulated in PDX5 versus PDX1, which significantly stratified patients according to survival. The genes UTRN, CELF2, and SST stratify patients with low survival when their expression is low, concordantly with our results on PDXs. We also detected genes with decreased expression in the PDXs that were associated, in this case with good prognosis, such as TAF10 and GSTP1. With these findings we have identified a set of genes, most of which were not previously associated with neuroblastoma, that seem to be relevant in the progression of the disease.

## 3. Discussion

In this manuscript, we explored the changes associated with the engraftment and growth of neuroblastoma patient tumor samples in recipient immunocompromised mice. We confirm that, when serially transplanted into mice, without external culture, up until the fifth passage, tumor xenografts become visible after shorter times, grow faster, and compromise host survival earlier. A necessary mechanism of adaptation to the new host environment after transplantation has been described for PDXs, in general, and for neuroblastoma, in particular [14,17]. This adaptation is normally attributed to the necessary change from a human stroma to a murine one. Since normally, PDXs are implanted in immunocompromised mice, this adaptation is not supposed to be greatly influenced by the immune setting, although the absence of an immune response and the progressive elimination of the human immune infiltrate might account for increased tumor growth. We observed some genes related to immune function decreasing along PDX passages, indicating that this might be a possibility (Appendix A). Adaptation is, presumably, largely triggered by a change in stroma and tumor microenvironment [21]. Unsurprisingly, many of the genes found differentially expressed in PDX passage 5 versus previous ones are related to extracellular matrix remodeling, including matrix components, such as laminins and collagens, or adhesion molecules, such as SEMA5A or CD44. Another described mechanism of adaptation involves metabolic changes in tumor cells [22]. Some of the genes identified as changing in PDX5 versus PDX1 are related to this molecular function, including, for the first time in this work, some genes that have been associated with survival in neuroblastoma patients.

The gene expression comparison performed between the different PDX passages reflects subtle but significant changes. Although we could observe changes in the expression of many genes, in most cases the fold change obtained was not greater than two. This probably reflects progressive changes leading to different tumor phenotypes with time. Monitoring protein expression by immunohistochemistry confirms this change in proliferation and differentiation character of tumors, with successive PDXs being progressively enriched in proliferative undifferentiated cells. Despite this modest change in expression levels, the number of genes identified is quite large, indicating a drift in gene expression patterns. Some of these changes could be observed over time with the STEM analysis, but most differences were better observed when comparing PDX1 to PDX5 directly. STEM gene expression showed some mixed patterns followed significantly by a selection of genes. This probably indicates temporal complex readaptation. More uniform up or down patterns probably indicate permanent gene expression changes provoked by constant selective pressure along passages, most probably due to the exposition to the host microenvironment. Genes following these progressive patterns are involved in neural differentiation, cell proliferation or matrix remodeling, probably leading the tumor to the final proliferative and undifferentiated phenotype. An alternative hypothesis is the existence of a more proliferative clone that is sequentially selected, taking over the whole population, although this possibility could not be tested with the approach taken [7].

To the best of our knowledge this is the first comprehensive analysis of miRNA expression in neuroblastoma PDX. Our approach has identified a total of 142 miRNAs differentially expressed in PDX5 versus PDX1. Among the ones changing most significantly are several involved in regulation of cell proliferation and differentiation. For example, miR429 regulates proliferation and invasion of endometrial, prostate, and breast carcinoma [23,24], and miR758 inhibits proliferation, migration, and invasion [25]. miR3167 is located on chromosome 11q and is important for neuroblastoma, but for unknown reasons [26]. Our interactome analysis has also identified miRNAs differentially changing by serial PDX passage concomitantly with some of their mRNA targets. Genes involved in neuroblastoma, such as MYCN or CD44, are among them. It is possible that the observed changes in mRNA are affected by changes in expression of effector miRNAs, but we cannot definitively confirm this case at this stage.

Given the increase of proliferative and undifferentiated character with time, we used the PDX5 versus PDX1 expression comparison approach to identify new modulators of neuroblastoma progression. We have described for the first time that a high expression of miRNA-99a stratifies patients with worse survival and correlates with MYCN expression. Interestingly, this association is independent of the neuroblastoma tumor stage, maybe indicating a direct connection with MYCN oncogene pathway. Targets for this miRNA in neuroblastoma are unknown to date. microRNA inhibitors have been described and are under development for translational use [27]

We have also identified new genes mediating survival in neuroblastoma patients. Several of them are involved in proliferation. Somatostatin is associated with a lower survival when downregulated and is found with a lower expression in PDX5. It has been described as affecting proliferation and inhibiting cell division in neuroblastoma [28]. SLC1A5 is a Rho GTPase effector involved in glutamine intake and ferroptosis, with important implications for cancer cell metabolism and neurological disorders, such as Parkinson’s disease [29,30]. Interestingly, several inhibitors for SLCA15 are under development for cancer treatment, especially for small-cell lung cancer [31]. Several agents upregulate utrophin (UTRN), a gene that we find downregulated in PDX5 versus PDX1 in our analysis, and are currently being tested for their use in the treatment of muscular dystrophy [32]. Most genes identified here that have an effect on neuroblastoma survival have not been explored previously for this disease. These findings demonstrate that the results obtained from PDX studies can be relevant for patients, as they are the best and most physiologically sound models for studying patient samples and human tumors in an in vivo setting. In this study, we used three different PDX models and performed a detailed expression and bioinformatics analysis of one of them. Ideally, the analysis could be extended to more PDX samples in later studies, but we have observed a similar behavior in all the PDXs developed in the laboratory, and the findings described here, along with the detailed analysis of PDX NB27T, have been extended in our work to studies in several patient cohorts with large numbers of human tumor samples with different characteristics, highlighting their relevance.

With this work we have demonstrated that adaptation of patient tumors as PDXs extends beyond temporal host-oriented changes and renders more proliferative and aggressive experimental tumors. This observation suggests a potential use of PDX as a model for neuroblastoma aggressiveness and not just a surrogate of the patient’s tumor. Neuroblastoma PDXs have been described as being more metastatic than conventional cell line-derived xenografts [10]. We observed changes in gene expression and phenotype concordant with this idea.

The data presented adds to the idea of PDXs as good models for studying neuroblastoma biology and behavior and expanding scarce patient tumor material. PDX models have also been demonstrated to be more useful than the transgenic neuroblastoma models available, which develop tumors that do not resemble the nature and behavior of human tumors. The expression changes observed between PDX passages were relevant but minor when looked at in the overall heatmap profiles. We conclude that the main characteristics were maintained between the samples and the tumor growing in mice. However, attention must be paid to the increase in aggressiveness and proliferative potential observed with serial passage in mice. We can use this feature to identify new genes involved in the adaptation of tumors in vivo that spontaneously convert into more aggressive forms without new external selective insults. With this approach, we have identified new miRNAs and genes with an influence on neuroblastoma patient survival. Future directions would be to identify target genes for miRNA-99a and to discover the roles of the genes identified in neuroblastoma biology and progression.

## 4. Materials and Methods

### 4.1. Patient-Derived Xenografts (PDXs)

Patient-derived xenografts (PDXs) were established from freshly obtained human stage 4 neuroblastoma tumor samples provided by the Biobank of the Virgen del Rocío Hospital in Seville. Relevant clinical information and characteristics are presented in Appendix A. A 3 × 3 mm-sized fragment of the tumor was immediately obtained and placed with 30 μL of cold undiluted Matrigel (Corning Matrigel Matrix Phenol Red-Free, LDEV-Free. Ref: 356237, Corning Inc. New York, NY, USA) on a subcutaneous pocket performed on C.B-17 SCID mice (Harlan Laboratories, Indianapolis, IN, USA). Engrafted tumors were maintained in mice and monitored until they reached a volume of 1 cm^3^, then the tumor corresponding to the first passage of PDX (PDX1) was collected. A sample of this PDX1 was fixed in 4% PFA and embedded in paraffin for subsequent immunohistochemical study. The remaining tumor tissue was minced into 3 × 3 mm-sized fragments and freshly transferred to new recipient mice to generate the second passage of PDX (PDX2). This process was repeated until 5 serial in vivo passages were achieved. Engrafted mice were routinely checked for tumors and other signs of malignancy at sacrifice.

### 4.2. RNA Extraction and mRNA and miRNA Array

The PDX tumors formed were collected, minced into small pieces, and immediately frozen in liquid nitrogen and stored at −80 °C. Once all the PDXs tumors (PDX1 to PDX5) were collected, individual tissue samples were lysed using QIAzol Lysis Reagent (Qiagen, Venlo, Netherlands) and homogenized. RNA extraction was performed using miRNeasy Mini Kit (Qiagen, cat. no. 217004) according to the recommended procedures, and DNA was digested using DNase I. Quantification was performed using a fluorometric QubitTM RNA-BR Assay Kit (Invitrogen, Thermo Fisher Scientific, Waltham, MA, USA) and RNA integrity was analyzed using a BioAnalyzer2100 (Agilent Technologies, Santa Clara, CA, USA). The RNA was amplified and labeled using a GeneChip WT PLUS Reagent kit (Affimetrix, Santa Clara, CA, USA), hybridized to Affymetrix Clariom D Assay human array (Affimetrix), washed, and stained according to the manufacturer’s instructions. Subsequently, for miRNA analysis, the RNA extracted from the PDXs was also amplified and labeled using a Flash Tag Biotin HSR RNA labeling kit (Affimetrix), hybridized to GeneChip miRNAs Human Arrays (Affimetrix). A custom expression array was created by means of removing unspecific probes (those that could potentially hybridize mouse cDNA)

### 4.3. Microarray Analysis

Using custom R (v3.5.0) scripts, arrays were then normalized with the robust multi-array average (RMA) method from *oligo* package, and differential gene expression analysis was performed with *limma* package. Annotation was built in base to Clariom_D_Human.na36.hg38.plus_HumanMouseXenograft.transcript annotation set. Mouse gene probes and non-annotated genome areas for these arrays were removed from the analysis. Graphical representation was performed by using *ComplexHeatmap*, *ggplot2* and *tydiverse* libraries.

Gene expression kinetics between PDX1 and PDX5 (0 to 4 steps, respectively) were traced and plotted by using Short Time-series Expression Miner Software (STEM) [18]. The clustering method was k-means from normalized data and the maximun number of profiles generated was 50. The significance level was 0.05 and the correction method used was Bonferroni. The clustering minimun value for correlation was 0.7. Gene ontology was performed by using STEM software. Representation was performed by using *ggplot2* in RStudio.

miRNA interactome analysis was carried out by using Kruskal–Wallis, and ANOVA analysis using STEM [18].

Cell population markers were obtained from prior publications [6,7,19,20]. CD44 (as MES marker) and CDC42 (as 1p36 loss marker) genes were also included.

### 4.4. Patient Data Correlation

Correlation with patient data samples was performed by using ‘R2: Genomics Analysis and Visualization Platform (http://r2.amc.nl (accessed on 15 June 2021))’ and GSE45547, GSE16476 and Tumor Neuroblastoma—Westermann—579—tpm—gencode19 data bases, according to INSS staging system, risk stratification, MYCN status, or death event.

### 4.5. Immunohistochemistry

For immunohistochemistry staining, paraffin-embedded sections (5 μm) were obtained from PDXs samples, hydrated, and treated for antigen retrieval with citrate 0.1 M (pH 6.0). Endogen peroxidase was inactivated with methanol, and samples were incubated with the corresponding antibodies. A Vectastain ABC kit and DAB peroxidase substrate kit (Vector Laboratories, Burlingame, CA, USA) were used following recommended procedures. Primary antibodies used were rabbit polyclonal anti VRK1 (Sigma-Aldrich/Merck KGaA, Darmstadt, ermany, #HPA000660; 1:500), rabbit anti Ki67 (Thermo Scientific, Waltham, MA, USA, #RM-9106; 1:500), and rabbit polyclonal anti DDC (Millipore/Merck KGaA, Darmstadt, Germany, #AB1569; 1:1000). Secondary antibodies used were biotinylated-conjugated anti-mouse and biotinylated-conjugated anti-rabbit (Vector Laboratories Burlingame, CA, USA, #BA-9200 and #BA-1000, respectively; 1:1000). 

## Figures and Tables

**Figure 1 ijms-24-01590-f001:**
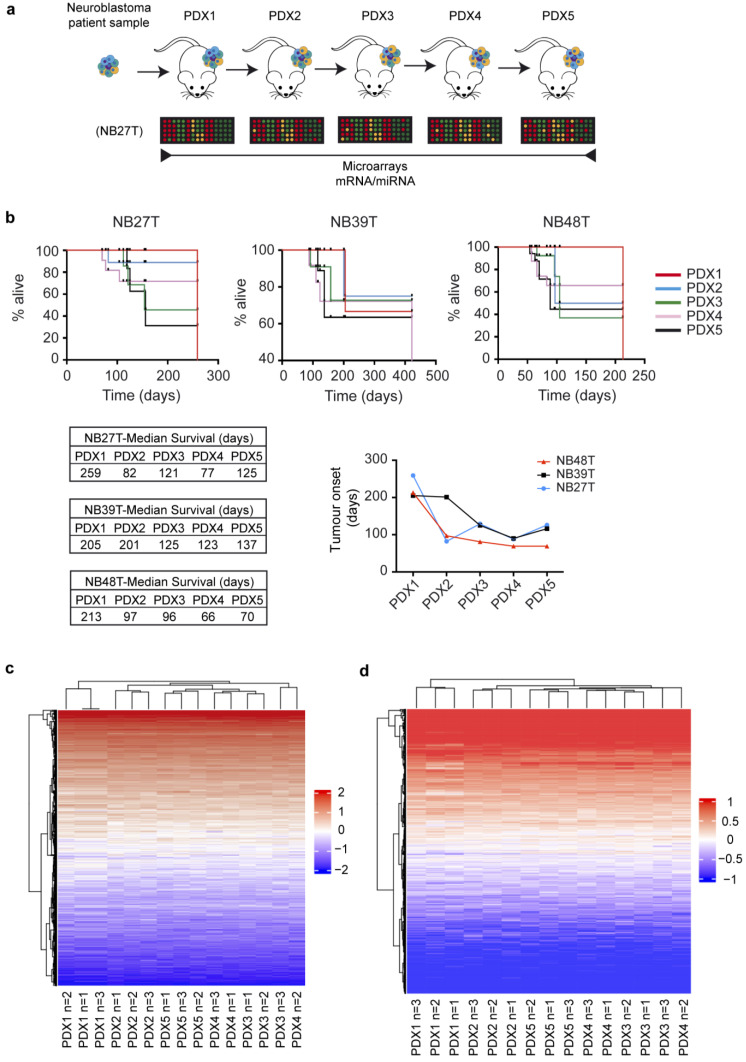
(**a**) Scheme showing PDX serial transplantation approach. (**b**) Percentage of mice alive at the indicated time points and tumor onset of different NB PDXs. Median survival is indicated. (**c**) Normalized summary heatmap with the mRNA gene expression changes in PDX NB27T from all passages and replicates. (**d**) Normalized summary heatmap with the miRNA expression changes in PDX NB27T from all passages and replicates.

**Figure 2 ijms-24-01590-f002:**
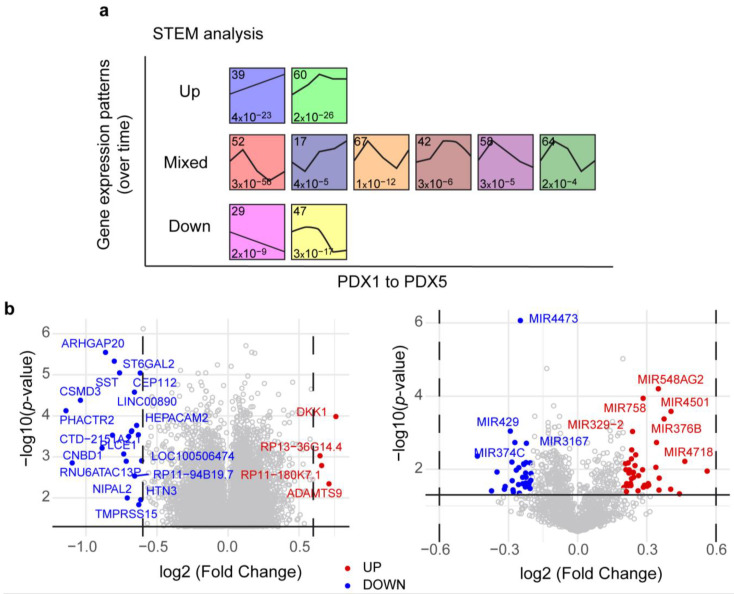
(**a**) Gene expression profiles obtained from the STEM analysis of PDX NB27T. In each box, the number in the top left is number of genes, in the bottom left, *p* value. Only the ones with significant *p* values are shown. Profiles are grouped as Up, Down, or Mixed. (**b**) Volcano plots showing the most significant mRNA expression changes (**left**) or miRNA changes (**right**) from the PDX5 vs PDX1 comparison. In blue, the downregulated genes in PDX5, and in red, the upregulated genes in PDX5.

**Figure 3 ijms-24-01590-f003:**
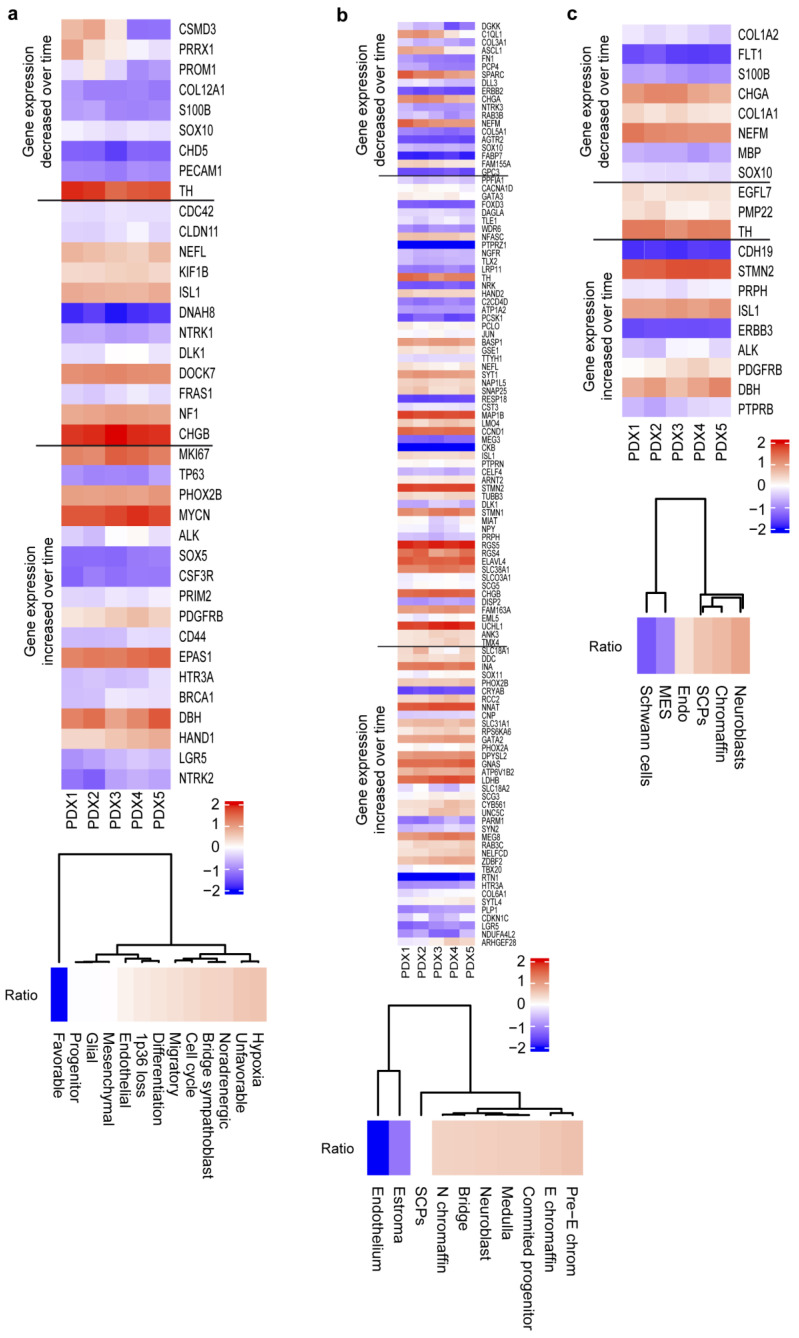
Heatmaps showing normalized gene expression changes on PDX passage 1 to 5 of PDX NB27T for genes on different published signatures. Ratio of the relative change in the different cell types is shown in each case: (**a**) signature from [6], (**b**) signature from [20], (**c**) signature from [7]. Lines delimitate genes decreasing or increasing their expression with time.

**Figure 4 ijms-24-01590-f004:**
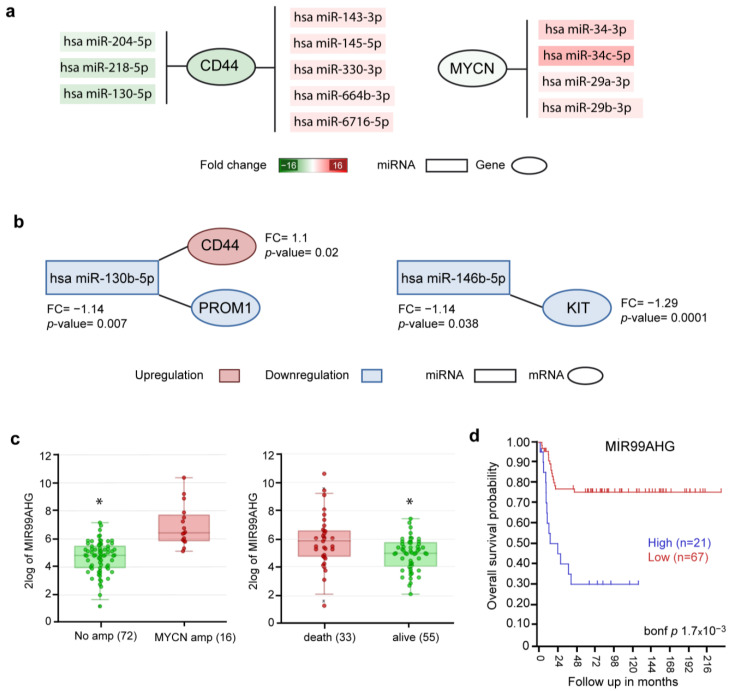
(**a**) miRNAs found differentially expressed among PDX passages affecting CD44 expression or MYCN expression. (**b**) miRNAs affecting CD44, PROM1, or KIT mRNAs. *p* values are shown. FC: fold change (**c**) expression of the miRNA99-a gene (MIR99AHG) in neuroblastoma patient tumor samples divided according to MYCN amplification or patient outcome. *: *p* < 0.005. (**d**) Kaplan– Meyer curves showing overall survival of patients according to miRNA-99a expression.

**Figure 5 ijms-24-01590-f005:**
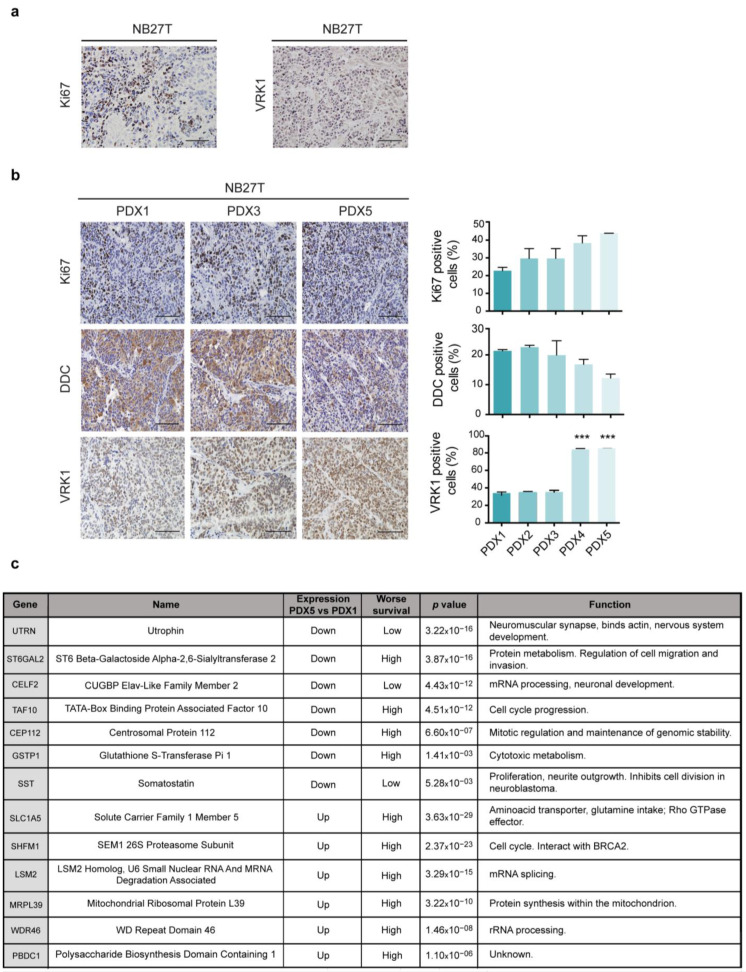
(**a**) Immunohistochemistry showing expression of the indicated markers in the original NB27T patient sample. Scale bar: 100 μm. (**b**) Immunohistochemistry showing expression of the indicated markers in PDX NB27T tissue from the indicated PDXs passages. Quantification of the expression is shown on the graphs. Scale bar: 100 μm. ***: *p* < 0.001 (**c**) Table showing genes differentially expressed between PDX5 and PDX1, whose expressions stratify neuroblastoma patients according to overall survival. Low: low expression, resulting in worse survival; high: high expression, resulting in worse survival.

## Data Availability

Publicly available neuroblastoma patient expression datasets GSE45547, GSE16476 and Tumor Neuroblastoma—Westermann—579—tpm—gencode19 were used and can be accessed via ‘R2: Genomics Analysis and Visualization Platform (http://r2.amc.nl (accessed on 15 June 2021))’.

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
