# Peer review of "Analysis of Serial Neuroblastoma PDX Passages in Mice Allows the Identification of New Mediators of Neuroblastoma Aggressiveness"

_ijms, 2023, doi:10.3390/ijms24021590_

Round 1

Reviewer 1 Report

The manuscript entitled "Analysis of serial neuroblastoma PDX passages in mice allows the identification of new mediators of neuroblastoma aggressiveness" by Maria et al. was presented as an original Article in which the authors analyzed consecutive PDX passages in mice at both transcriptomic and histological levels. Authors have studied temporal changes in mRNA and miRNA expression and correlated those with neuroblastoma aggressiveness in patient data. They have also observed a shortening in tumor onset time with an increase in proliferative potential in the PDXs over the serial passage. I believe the authors have shown some good results, and the paper would interest a broad community working on neuroblastoma cancer. However, the manuscript is still preliminary before approving it for publication.

 Major points:

1.     The manuscript needs restructuring in the writing part with lots of language corrections as it contains many typos, which makes the manuscript hard to understand. A few typos are mentioned here: Line 79 and 106 has typos, "Sucesive, agresiveness and ad."

2.     What was the engraftment rate for tumors in NOD/SCID mice, as these mice have less immunodeficiency than NSG/NRG mice?

3.     It is reported that aging immunodeficient mouse strains are susceptible to spontaneous murine lymphomas. Did the authors examine the presence of lymphoma, including splenomegaly and enlarged liver and lymph nodes?

4.     Was the Matrigel used in the engraftment experiment bought commercially or synthesized in the lab?

5.     Is the tumor's aggressiveness over several passages due to the elimination/decrease of the human immune infiltrate?

Minor:

1.     Please consider adding IHC data from two different patients.

2.     The representation of data in figures could be improved.

Reviewer 2 Report

Gomez-Muñoz et al describe a strategy to identify new mediators of neuroblastoma aggressiveness using patient-derived xenograft (PDX) models. They have compared gene expression profiles and miRNA analysis of the same PDX model at different passages. The authors have identified genes such as UTRN, CELF2 and SST that could be associated with low survival in patients. The issue is of interest among pediatric oncologists because of the emerging role of individualized molecular cancer therapies.

The study is comprehensive and the methods appropriate. The English scientific language can be improved. The patient/tumor numbers are far too small for confirmatory statistics as only three PDX have been included in these study (NB27T, NB39T, NB48T), and the results are not shown for all of them. Thus, it is difficult that the conclusions presented in this article could be expanded to Neuroblastoma tumors in general and not a specific characteristic of the PDX model analyzed. More PDX models should be included in this study.

Abstract:

The wording of the abstract could be improved.

Introduction:

It should be improve in order to male clear the aim of the study.

Results:

Figure 1b. Why the different mice are sacrificed at different time points between NB27T, NB39T and NB48T? Specify median survivals in a table. Do you have any data regarding why the tumors need such a prolonged time to engraft and growth? Which volume is established as tumor endpoint?

Figure 1c. I do not understand PDX 1-2, 1-1, 1-3, etc. Please clarify. Are from the same PDX different passages? Are from different PDXs? Specify which PDX you are analyzing in the figure.

Supp Table 1. Include more information about patients: age at diagnosis, metastasis at diagnosis, treatment if any, molecular characteristics and histological characteristics of the tumor patients. The results observed could be partially produced depending on the treatment received by patients (clone selection, etc.).

Figure 2 and 3. Specify which PDX are you analyzing.

Figure 4. Figure c and d are not specified in the text. No information a part from pathway is shown. Please improve.

Figure 5a. Include patient histology in order to compare.

Figure 5b. Information about the gene expression from the patient should be included.

Methods:

Why is necessary the use of Matrigel if neuroblastoma is a soft-tissue tumor?

All the passages have been done in CB17 SCID?

The tumors where implanted from fresh samples? If not, specify the protocol used.

Include patient information,

Survival correlations should be performed from all the genes and miRNAs analyzed. Include a supplementary figure.

Discussion

How can be your data obtained from PDX model be translated directly to patients?

How can we conclude that the results could be clinically relevant if a low number of samples have been analyzed?

How can we conclude that PDX models are good models to study neuroblastoma if the tumor characteristics change between passages?

Are there any treatments or potential treatment that target any of the genes and miRNA that you have identified as mediators of neuroblastoma aggressiveness

I think the points mentioned above need to be discussed along with other limitations of the study.

Round 2

Reviewer 1 Report

The manuscript has substantially improved after the revision. The authors have discussed and explained the significant concerns. I have a minor question did authors examine the Matrigel for possible murine viral contamination, and did they use the same batch of Matrigel for all five serial PDX engrafts?

Author Response

I have a minor question did authors examine the Matrigel for possible murine viral contamination, and did they use the same batch of Matrigel for all five serial PDX engrafts?

The commercial Matrigel provided by Corning is routinely tested for a number of pathogens, including virus, and other contaminants. It is LDEV-free, a common viral contaminant in this kind of preparations. Every batch comes with the results of the analysis performed by the manufacturer, so we believe viral murine contamination is not an issue. In this case we used different aliquots from the same Matrigel batch for our experiments.